# Circulating Cell-Free Tumour DNA for Early Detection of Pancreatic Cancer

**DOI:** 10.3390/cancers12123704

**Published:** 2020-12-09

**Authors:** Jedrzej J. Jaworski, Robert D. Morgan, Shivan Sivakumar

**Affiliations:** 1MRC Laboratory of Molecular Biology, Francis Crick Avenue, Cambridge CB2 0QH, UK; jjaworski@mrc-lmb.cam.ac.uk; 2Department of Medical Oncology, Christie NHS Foundation Trust, Manchester M20 4BX, UK; robert.morgan@christie.nhs.uk; 3Division of Cancer Sciences, Faculty of Biology, Medicine and Health, University of Manchester, Manchester M13 9PL, UK; 4Department of Oncology, University of Oxford, Old Road Campus Research Building, Roosevelt Drive, Oxford OX3 7DQ, UK; 5Department of Medical Oncology, Oxford University Hospitals NHS Foundation Trust, Oxford OX3 7LE, UK

**Keywords:** circulating cell-free tumour DNA, pancreatic cancer, pancreatic ductal adenocarcinoma

## Abstract

**Simple Summary:**

Pancreatic ductal adenocarcinoma (PDAC) has a poor 5-year survival rate and is the 7th leading cause of cancer-related deaths in the world. The high mortality for this disease is partly due to late presentation rendering therapeutics ineffective. Since a majority of patients are diagnosed at advanced stages due to the lack of specific symptoms, and the prognosis is linked to the stage of detection, there is a need for robust early detection methods for PDAC. Here, we review the potential use of circulating tumour DNA (ctDNA) as a non-invasive biomarker in the early detection of pancreatic cancer. In brief ctDNA levels in blood correlate with disease progression but its widespread application for early PDAC detection requires further investigation and potentially, a combination of ctDNA sequence and methylome analysis with other, protein-based biomarkers.

**Abstract:**

Pancreatic cancer is a lethal disease, with mortality rates negatively associated with the stage at which the disease is detected. Early detection is therefore critical to improving survival outcomes. A recent focus of research for early detection is the use of circulating cell-free tumour DNA (ctDNA). The detection of ctDNA offers potential as a relatively non-invasive method of diagnosing pancreatic cancer by using genetic sequencing technology to detect tumour-specific mutational signatures in blood samples before symptoms manifest. These technologies are limited by a number of factors that lower sensitivity and specificity, including low levels of detectable ctDNA in early stage disease and contamination with non-cancer circulating cell-free DNA. However, genetic and epigenetic analysis of ctDNA in combination with other standard diagnostic tests may improve early detection rates. In this review, we evaluate the genetic and epigenetic methods under investigation in diagnosing pancreatic cancer and provide a perspective for future developments.

## 1. Introduction

Pancreatic ductal adenocarcinoma (PDAC) is the 14th most prevalent cancer and the seventh most common cause of cancer-related death worldwide [1]. Its incidence is expected to double by 2030, becoming the second-leading cause of cancer-related death in the United States [2,3]. Pancreatic ductal adenocarcinoma also has the lowest five-year survival rate of any malignancy, which is around 5–10% [4,5]. This is attributed to the disease frequently presenting and/or being detected in an advanced stage, in which a cure is highly unlikely [6]. Indeed, a potential cure for PDAC involves surgical resection with adjuvant chemotherapy [7]. Alternatively, borderline resectable and locally advanced PDAC cases may benefit from neoadjuvant therapy, which can be safer than surgery. A number of studies have compared the two approaches [8,9,10,11,12,13], but the broad clinical utility of neoadjuvant therapy still requires further investigation in a phase 3 clinical trial setting [14]. Altogether, approximately 85% of patients diagnosed with PDAC are not suitable for potentially curative therapy due to locally advanced or metastatic spread. Early detection of PDAC is thus critical to allow access to surgery and therefore, for the improvement in survival outcomes for patients. One area of current research is using liquid biopsies to detect the disease at an early stage.

The development of next-generation DNA sequencing methods, together with technical advances in molecular biology, has sparked interest in liquid biopsy as a tool for the early detection of cancer [15]. Currently, the only blood-based marker used in the diagnosis of pancreatic cancer is serum CA19.9 [16]; however, its relatively low sensitivity (79%) and specificity (82%) limits its validity as a stand-alone test [17,18]. Newly uncovered circulating biomarkers raise hopes for more sensitive and specific diagnostic tests that can analyse a myriad of body fluids e.g., saliva, blood or urine [19,20,21]. Such methods include the detection of exosomes, DNA, RNA or circulating tumour cells (CTCs) (Figure 1). The numbers of detectable CTCs are often low in many tumour types, particularly in early stage disease, and there are no validated biomarkers for cell selection, making this approach somewhat limited [22]. In contrast, circulating nucleic acids are more prevalent in early stage disease, offering a potential biological marker for detection. In this review, we discuss the application of circulating tumour DNA (ctDNA) in the early detection of pancreatic cancer.

## 2. Liquid Biopsies for ctDNA

A liquid biopsy is a non-invasive investigation that holds great promise in detecting early stage cancer. It involves sampling blood and/or other body fluids for the presence of ctDNA or CTCs. Initial research has focused on advanced cancers, where ctDNA levels are highest and therefore isolation and sequencing of tumour-specific nucleic acids are more feasible [23,24]. Indeed, in advanced stage pancreatic cancer, the presence of ctDNA in the blood has been associated with both relapse and residual disease following surgery [25,26], and could be used to tailor therapeutic approaches [27].

Liquid biopsies are more accessible than tumour biopsies, and can provide a new diagnostic tool, particularly in tumour types where invasive biopsies are comparatively more unsafe and difficult. In addition, while single-site tumour biopsies may not reflect tumour heterogeneity, particularly in metastatic disease, ctDNA analysis offers the potential to provide a method to detect inter- and intra-tumour heterogeneity across metastatic sites [28]. Therefore, ctDNA analysis offers the potential to detect early stage PDAC and reduce the number of potentially harmful procedures, as well as limiting costly and unnecessary therapies that may be associated with toxicity [29].

Circulating cell-free DNA in blood was first described by Mandel and Metais in 1948 [30]. Subsequent investigations led to the observation in 1977 by Leon et al. [31] that circulating cell-free DNA (cfDNA) levels are increased in cancer. Further studies have since indicated that cfDNA levels are also elevated in other pathophysiological processes, such as following physical trauma [32], cerebral infarction [33], physical exercise [34] and solid organ transplantation [35]. Another major breakthrough came from the discovery of the presence of foetal Y-chromosomal cfDNA in maternal blood [36], which led to the development of non-invasive prenatal testing [37]; a screening method now used to detect congenital chromosomal abnormalities [38,39,40]. In the context of cancer medicine, cfDNA studies have shown that the fraction of the cfDNA produced directly from cancer cells (i.e., ctDNA) can also be detected, but has a shorter half-life than non-cancerous cfDNA [41] and therefore requires almost immediate sample processing.

Circulating cell-free DNA is fragmented, typically double-stranded and approximately 150–350 base pairs in length [42]. It can be released from a variety of healthy cells, but the majority originates from haematopoietic cells [43,44]. In patients diagnosed with cancer, cfDNA is also released from cancer cells, though the mechanism of ctDNA release remains unclear—either active secretion and/or release in cell death, supported by studies showing that ctDNA levels increase rapidly after treatment is administered in a range of cancers [45,46,47]. The mechanism of DNA release might affect cancer detection, as actively secreted ctDNA may be biased in certain chemotherapy-sensitive metastatic sites. ctDNA constitutes between approximately 0.1% and >90% of all cfDNA depending on a number of factors, in particular cancer stage [24,48]. Moreover, Bettegowda et al. showed that early stage cancers have a lower percentage of ctDNA then late stage malignancies [24]. For pancreatic cancer, ctDNA was detected in 48% of patients with localised tumours, but in more than 80% of advanced cancers. This shows a potential limitation of ctDNA as an early detection tool for pancreatic cancer.

## 3. Detection of ctDNA

An important property of ctDNA related to early cancer detection is its low stability; different studies estimate ctDNA half-life to be between 16 and 150 min [48,49,50], thus necessitating longitudinal analysis of tumour responses to therapy and/or disease progression. This short half-life also brings a technical challenge: the instability of cfDNA combined with the propensity for blood cells to lyse once outside the body, resulting in contamination, necessitates cfDNA separation by centrifugation within no more than 4 h of sample collection in standard phlebotomy tubes [15,51,52], which creates logistic burden and does not allow for central processing in specialised centres, therefore limiting widespread clinical implementation. In an attempt to address this issue, recent developments allow for the ability to fix and stabilise cfDNA for 7–14 days at room temperature [53]. Specifically, Cell-Free DNA BCT^®^ (Streck, La Vista, NV, USA) tubes are estimated to preserve cfDNA in blood for 7 days due to an improved vacuum and moisture retention system [54,55,56] and have been used in a number of recent diagnostic studies of different tumour types [57,58].

Another significant challenge in the diagnostic utility of ctDNA is its low concentration in plasma, complicating detection and analysis. There is approximately 10–15 ng of cfDNA per millilitre and ctDNA constitutes a small fraction of cfDNA in most early stage of cancers [48]. These factors highlight the need for ultrasensitive detection techniques. In clinical trials, current sequencing methods involve targeted or genome-wide approaches to genetic sequencing. The former is more prevalent and only detects hotspot mutations in specific genes. This technology could be applicable to somatic *KRAS* mutations ubiquitously found in PDAC. Conversely, genome-wide sequencing assays are an unrestricted method that is more expensive and technically challenging, but identifies many more mutations across the genome as well as structural variants and mutational signatures. Currently, the most sensitive methods are polymerase chain reaction (PCR)–based approaches, including single-molecule PCR BEAMing [59], TAm-Seq [60], digital PCR [61] and droplet digital PCR [62]. Next-generation sequencing techniques are also widely used, with the biggest limiting factor being the low fidelity of DNA polymerase [60,63,64]. However, the incorporation of deep sequencing, molecular barcoding and digital error suppression can increase sensitivity and specificity of the approach [65,66].

## 4. ctDNA in Pancreatic Cancer

Initial insights into the utility of ctDNA in the diagnosis of pancreatic cancer were elucidated by Shapiro et al. [67], who showed that ctDNA can be found in patients diagnosed with pancreatic cancer, but is absent in healthy individuals. Almost 10 years after these initial findings, Sorenson et al. [68] were able to detect somatic *KRAS* mutations in the plasma of patients diagnosed with pancreatic cancer. These mutations matched those found in the tumour, sparking interest in ctDNA as an early detection marker for pancreatic cancer. Indeed, *KRAS* is a candidate gene for early detection because it is considered a clonal oncogenic driver of pancreatic cancer and is present in ~95% of cases [69,70]. A description of the most relevant studies of ctDNA application in PDAC detection and monitoring is provided in Table 1.

## 5. The Use of ctDNA to Detect Early Stage Pancreatic Cancer

Despite the promising results emerging from ctDNA use, the clinical applicability of liquid biopsy remains debatable across tumour types. A number of studies have shown low sensitivity and specificity of ctDNA application, particularly in breast [85] and lung [86] early stage cancers. Similarly, in a large case–control study, Calvez-Kelm et al. [76] reported that ctDNA is a less accurate method to detect early stage PDAC than CA19–9, and a combination of the two tests did not increase sensitivity. This finding was supported by a more recent study by Fiala et al. [87], which showed that the current sequencing technology was unable to detect ctDNA of pancreatic cancer less than 10 mm in diameter with detection efficiency increasing with cancer progression.

As a result, ctDNA analysis for early detection is currently unable to be used as a stand-alone method to detect early stage cancer [24]. Therefore, various approaches of combining ctDNA with other molecular biomarkers are being evaluated. Several studies have reported the applicability of such approaches in PDAC. Indeed, CA19–9 was shown to be a less sensitive marker than when combined with TIMP-1 protein testing [88]. In another study, Capello et al. showed that the sensitivity of PDAC detection is even higher when the two factors are combined with LRG-1 protein testing [89]. In an attempt to integrate protein biomarkers and ctDNA in PDAC detection, Cohen et al. [78] presented a study incorporating the data from *KRAS* ctDNA sequencing and four additional protein markers (CA19–9, CEA, HGF and OPN) and reported an increased specificity and sensitivity in early stage pancreatic cancer detection.

A similar method was exploited by Cohen et al. [80] in 2018. This group used CancerSEEK technology, based on ultra-deep PCR-based sequencing, to analyse 1933 distinct genomic loci combined with eight protein biomarkers. Based on data from liquid biopsies taken from 1005 patients with early stage, resectable cancer and 812 healthy individuals, the sensitivity rate of detection varied from 43% in stage I, 72% in stage II to 78% in stage III with specificity exceeding 99% (7 out of 812 controls were identified as cancer-positive). The sensitivity also varied based on tumour type e.g., 33% in breast cancer, 98% in ovarian cancer and 72% in pancreatic cancer. However, the accuracy of the approach has been questioned. In particular, Young et al. [90] criticised the use of case–control data where subject selection and the choice of a robust control are challenging. Specifically, no advanced stage cancers were included, and the healthy controls were likely to be biased and not represent the potential screening population (e.g., individuals with gastrointestinal tract symptoms or those characterised by genetic predispositions). In addition, the cases analysed included eight different cancer types of various stages, and it may therefore not be possible to draw robust conclusions about the utility of this approach in a single tumour type. For these reasons, the sensitivity and specificity reported could be biased and an additional study is required to validate the method.

One challenge related to ctDNA biology is a lack of consistency in the study design. As investigated tumour stages vary between studies, which include both case–control and cohort studies, a robust comparison of different methods is challenging. Moreover, PDAC should be confirmed using established techniques (e.g., histology), with a follow-up period for the detection of false-negative early stage cancers. In addition, controls should consist of individuals in high-risk groups, likely to be frequently screened, as well as healthy volunteers. If possible, calculations of sensitivity and specificity should be applied to each tumour type and stage separately to enable reliable comparison between studies.

Efficient, early diagnosis of PDAC also requires the identification of a screening population at risk of developing PDAC. There is a number of risk factors associated with PDAC, with the most relevant being new-onset diabetes mellitus [91], chronic pancreatitis [92] or genetic predisposition [93]. Ben et al. showed that, while long-lasting diabetes mellitus increases the risk of developing PDAC by 1.5-fold, recent detection (within 1–3 years from the disease onset) increases this ratio to 5–8-fold [94]. Similarly, the risk of developing PDAC increases eightfold over the five-year period following chronic pancreatitis diagnosis [92]. Finally, genetic predispositions are estimated to contribute to 5–10% of all PDAC cases [93], with mutations in the six most significant genes being reported in 5.5% of patients [95]. Altogether, the currently available data point at certain risk factors, which should determine the screening population for PDAC using ctDNA-based liquid biopsy.

## 6. Methylation Analysis in Cancer Diagnosis Based on Liquid Biopsy

Genome-wide epigenetic changes are common in cancer. Somatic methylations can be cancer-driving events and may precede genetic mutations during carcinogenesis [96,97,98]. As a result, epigenetics has attracted considerable research attention over the recent years. Introducing this property into ctDNA testing brings hope to increase the clinical utility of liquid biopsies [29].

Methylations of CpG islands provide a method to regulate gene expression by modulating transcription. The differences in methylation occupancy underlie many cancer types, with tumour suppressor genes being most frequently affected [99]. Certain methylations are tumour specific and therefore could be used to detect cancer as well as its tissue of origin based on ctDNA analysis. For example, *HOXA9* is methylated in 95% of high-grade ovarian serous carcinomas and together with *EN1* methylation levels can be used to distinguish between benign and malignant ovarian tumours with up to 98.8% sensitivity and 91.7% specificity [100].

Methylation assays for cfDNA have also been shown to be useful for PDAC detection and progression-tracking. Henriksen et al. [101] demonstrated that cancer-specific promoter methylation can be a marker for early PDAC detection. Moreover, Ligget et al. [102] showed that methylation analysis of 17 gene promoters in ctDNA was able to distinguish between patients diagnosed with chronic pancreatitis and PDAC with a sensitivity of 91.2% and specificity of 90.8%. More recently, Eissa et al. [103] identified the methylation of *ADAMTS1* and *BNC1* in ctDNA as a diagnostic biomarker in PDAC. Indeed, for localised PDAC, the sensitivity of this two-gene panel ctDNA analysis was 94.8%, and the specificity was 91.6%. These findings show promise in the early detection of PDAC and may enable differentiation from chronic pancreatitis.

A larger and more complex study was performed by Liu et al. [58], who investigated ctDNA application in early cancer detection in 15,254 subjects with or without over 50 types of cancer. The most relevant methylation regions were identified using whole-genome methylation assays followed by a wide-range analysis of the cancer detection properties. Subsequently, a targeted methylation assay was tested on a separate group of patients of all cancer stages. Specifically, stage I PDAC was detected with 63% sensitivity and 99% specificity, rising to 83% and 99%, respectively, in stage II. Taken together, these data provide further evidence that targeted ctDNA methylation analysis could be used in the early detection of PDAC.

Despite the improvements in sequencing technology, the low ctDNA concentration in patients with early stage disease is still one of the most significant challenges to the introduction of methylation analysis of ctDNA at the initial stage of cancer. So far, various attempts to unravel the methylome of ctDNA for cancer detection or monitoring have been done with whole-genome bisulphite sequencing [44,104]. However, this method is limited by input DNA degradation during bisulfide conversion [105], particularly in early cancer detection when a large enough blood sample would be required (with standard intake of up to 80 mL of whole blood) and ctDNA can constitute less than 0.1% of the cfDNA fraction [28]. Moreover, the methylome reflects both normal tissue and cancer heterogeneity and its analysis is thus complex, requiring a substantial amount of comparative data. Therefore, new methods are being developed to increase the efficiency of this technique. In 2018, Shen et al. [106] demonstrated a novel approach based on targeted amplification of cancer-specific methylation regions using *Enterobacter* phage lambda DNA to increase the initial DNA amount to at least 100 ng, increasing detection efficiency. Although this method demonstrates improved ctDNA methylome detection in a cost-effective manner, there is little knowledge of the cancer-specific DNA methylome, so a genome-wide methylation analysis appears to be more suitable.

## 7. Technical Advances Facilitate the Detection and Analysis of ctDNA and Also Unravel Its Properties

The discovery of ctDNA being typically shorter than cfDNA could be used for ctDNA isolation by size. The methylation pattern could also be exploited to detect cancer, instead of screening for specific methylations. Cancer progression is linked to genome-wide methylation loss, along with an accumulation of methyl groups in the regulatory regions of certain genes [107]. This property has been exploited by Sina et al. [108] who developed a promising method for cancer detection based on Methylscape analysis, leading to the increased aggregation of ctDNA on gold beads. Despite the low specificity observed and small sample size, the study presented a novel approach to ctDNA analysis. A description of the most relevant methylation studies of ctDNA application in the early PDAC detection is provided in Table 2.

## 8. Future Directions

Increasing understanding of cancer pathology and the introduction of early detection as well as targeted and immuno-therapies have revolutionised cancer care. Despite these advances, cancer is still the third leading cause of death in the world. Improving the detection of early stage disease is a promising strategy to advance survival outcomes [109]. ctDNA is a promising diagnostic biomarker, but does not yet represent a validated method for cancer detection. Newer approaches incorporate additional properties of ctDNA beyond sequencing, such as the length of fragments or methylation signatures, combined with cancer-associated protein biomarkers. This integrative approach is continuously revised and improved following technological advancement. Hence, ctDNA analysis has the potential to be introduced as a screening tool to detect early stage cancer, particularly in high-risk groups.

## 9. Conclusions

Circulating cell-free DNA analysis has a potential role in a number of clinical scenarios, including early cancer detection. However, studies on the applicability of ctDNA testing in clinical practice have had mixed outcomes and few ctDNA assays have so far demonstrated promise in the diagnosis of pancreatic cancer. The incorporation of additional markers and methylome analysis has been shown to improve the specificity and sensitivity of ctDNA analysis. Further evaluation is required before the introduction of ctDNA into pancreatic cancer detection and monitoring in clinics.

## Figures and Tables

**Figure 1 cancers-12-03704-f001:**
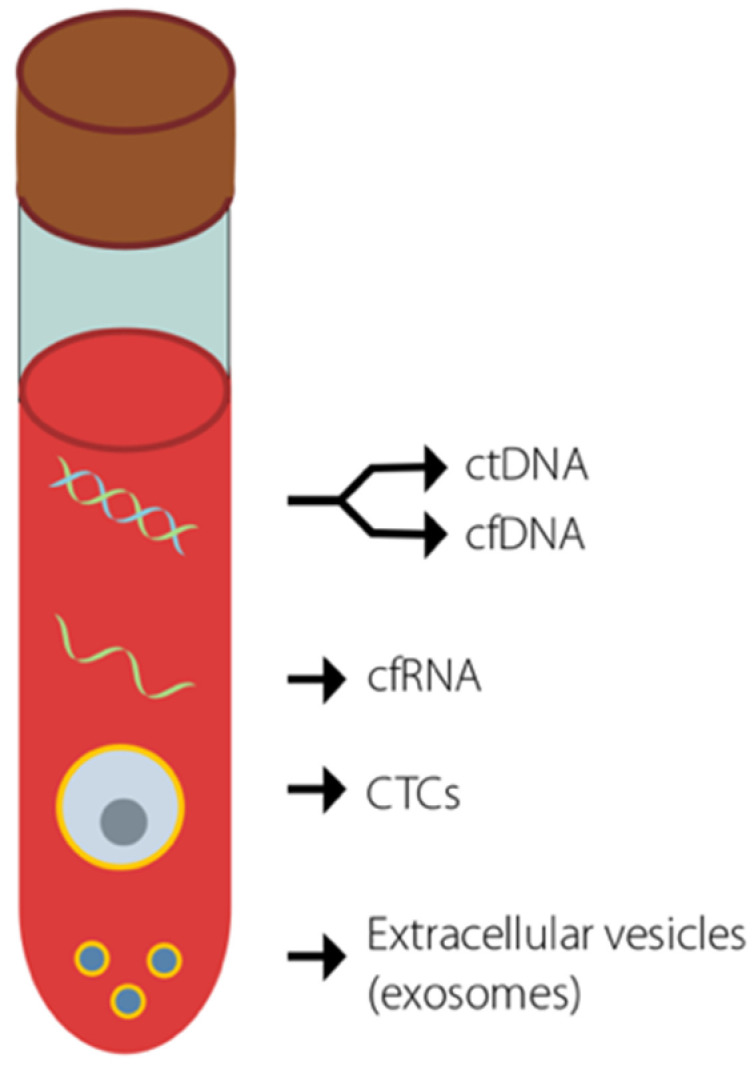
Potential cancer biomarkers present in the blood and detectable using liquid biopsies. Key biomarkers that are currently used in an attempt to detect early stage cancer are shown.

**Table 1 cancers-12-03704-t001:** Clinical studies investigating ctDNA in pancreatic cancer.

Author	Cancer Type	Detection Method	Interpretation	Number of Patients (*n*)	Sensitivity	Specificity	Diagnostic Targets
Sorenson et al., 1994 [68]	Different stages of PDAC	PCR	*KRAS* mutation found in blood of patients with PC, the mutations were identical with those in tumour biopsy	N/A	N/A	N/A	N/A
Maire et al., 2002 [71]	Late stages of PC	PCR	*KRAS2* mutation detected in 47% patients with PC, and in 13% of chronic pancreatitis; analysis of *KRAS2* + CA19.9 increased sensitivity to 98% and specificity to 77%	47 PC, 31 ctrl	47%/98% with CA19–9	13% in ctrl (specificity of 77% with CA19–9)	*KRAS2*
Dianxu et al., 2002 [72]	Early stages of PC	PCR-RFLP	*KRAS* mutation found in 70.7% of patients with PC but not in 21 controls. Combined with CA19–9, proportion increased to 90.2%	58 + 21 ctrl	70.7%/90.2% with CA19–9	N/A	*KRAS*/*KRAS* + CA19–9
Singh et al., 2015 [20]	All stages of PC	PCR-RFLP	ctDNA exceeding 62 ng/mL linked to lower OS and metastasis	N/A	N/A	N/A	N/A
Sausen et al., 2015 [25]	Stage II of PC	NGS, ddPCR	ctDNA detected in 43% of patients, ctDNA linked to an adverse prognosis and predicted relapse 6,5 months before CT detection	whole-exome analyses of 24 tumours, targeted genomic analyses of 77	N/A	N/A	N/A
Tjensvoll et al., 2016 [73]	All stages of PC	PNA-clamp PCR	*KRAS* mutation detected in 71% patients, ctDNA corresponded to CT results and CA19–9 levels	14 patients with several blood samples	71%	N/A	*KRAS* mutations
Berger et al., 2016 [74]	Metastatic PC	ddPCR	*KRAS* mutation in 41.7% of patients, 0% in control population	21 IPMN patients; 38 controls; 24 metastatic PDAC, 26 resected SCA; 16 borderline IPMN	41.7%	84.2%	*KRAS*^G12.D^ and *KRAS*^G12.V^
Brychta et al., 2016 * [75]	Early stage pancreatic cancer (mostly stage I & II)	ChIP-based digital PCR	Detection rates varied between 0% and 50% for specific mutations. *KRAS* mutation not detected in healthy patients	50 (82% of stage I & II)	72% (based on both liquid and standard biopsy)	N/A	*KRAS*^G12.D^, *KRAS*^G12.V^, and *KRAS*^G12.C^ mutations in blood and tumour samples
Le Calvez-Kelm et al., 2016 * [76]	All stages of PC	NGS	*KRAS* cfDNA mutations detected in 21.1% of cancers. No improvement over CA19–9	437 PC cases, 141 chronic pancreatitis subjects, 394 healthy controls	21.1%	96.3%	*KRAS*, CA 19–9
Takai et al., 2016 * [77]	All stages of PC	ddPCR and NGS	ddPCR detected KRAS mutation in 58.9% of non-resectable PC	259	58.8% in inoperable tumours	N/A	*KRAS* mutations
Cohen et al., 2017 * [78]	Resectable PDAC	PCR-based test and protein biomarkers	*KRAS* mutation detected in 30% of PC patients (66/221), in 66% when combined with protein biomarkers	221 with resectable PC, 182 controls	30% (only *KRAS*), 66%: *KRAS* + four protein biomarkers	N/A	*KRAS* mutations and protein biomarkers
Cheng et al., 2017 [79]	Metastatic PC	ddPCR, NGS	72.3% of PC patients presented with ctDNA-detected *KRAS* mutation	10: exome seq, 188 ddPCR,	76.9%	N/A	60 genes screened
Pietrasz et al., 2017 [26]	All stages of PC	NGS, ddPCR	ctDNA found in 48% of patients with PC, the presence of ctDNA was a predictor of an adverse prognosis	135, 31 resectable	48%	N/A	N/A
Cohen et al., 2018 * [80]	I–III stages of PC	NGS	Highly efficient multi-analyte test used	1005 patients of 8 cancer types, stage I–III	76% (any stage)	99%	CancerSEEK tested for 8 cancer types; 8 protein biomarkers and mutations in 1933 distinct genomic positions
Chen et al., 2018 [81]	All stages of PC	Meta-analysis of literature; significant in predicting OS and PFS	The presence of ctDNA or elevated cfDNA linked to poor prognosis	1243 from 18 articles	N/A	N/A	N/A
Bernard et al., 2019 [82]	Localised or metastatic PC	ddPCR from ctDNA and exosomes	ctDNA showed no correlation with outcomes, as opposed to exosome levels. However, detection of ctDNA post-resection correlated with lower PFS and OS	194 (overall receiving treatment, also with metastasis), 34 with resectable PC	N/A	N/A	N/A
Berger et al., 2019 * [83]	Resectable PC	Fluorimetry (HS Assay for cfDNA quantification)	CA19–9, THBS2 and cfDNA levels in combination were a better PC biomarker (c-statistics 0.90) than any of those separately	52	90%	N/A	thrombospondin-2 (THBS2), CA19–9
Liu et al., 2019 * [84]	Mostly stage I and II	hybrid-capture-based cfDNA sequencing (SLHC-seq)	ctDNA fragmentation pattern may affect the detection of early PC; cancer-specific mutations found in 88% patients, *KRAS* hotspots in 70%	112	88%	8 mutations detected in 28 heathy controls	791 cancer-specific mutations

* Early PDAC detection study. CA19–9, carbohydrate antigen 19–9; cfDNA, cell-free DNA; ctDNA, circulating tumour DNA; ctrl, control; ddPCR, digital droplet PCR; IPMN, intraductal papillary mucinous neoplasms; NGS, new-generation sequencing; OS, overall survival; PC, pancreatic cancer; PCR, polymerase chain reaction; PCR-RFLP, PCR Restriction Fragment Length Polymorphism; PDAC, pancreatic ductal adenocarcinoma; PFS, progression-free survival; SCA, serous cystadenoma.

**Table 2 cancers-12-03704-t002:** Selected methylation studies of ctDNA in pancreatic cancer.

Author	Cancer Type	Detection Method	Interpretation	Number of Patients (*n*)	Sensitivity	Specificity	Diagnostic Targets
Liggett et al., 2010 [102]	All stages of PC, chronic pancreatitis	Methylation-specific PCR	17 gene promoters were indicated as informative to differentiate between chronic pancreatitis and PC	30 overall	91.2%	90.8%	56 fragments in each sample (MethDet56)
Henriksen et al., 2016 [101]	All stages of PC	Methylation-specific PCR	The number of tested methylated genes was significantly (*p* < 0.001) higher in cancer patients than in control	97 PDAC	76%	83%	28 gene panel
Eissa et al., 2019 [103]	All stages of PC	Methylation on beads	Methylation of *ADAMTS1* and *BNC1* is a reliable marker for early detection of PC	39	97.3%	91.6%	*ADAMTS1* and *BNC1*
Liu et al., 2020 [58]	All stages, data shown here for stage I	Bisulfite sequencing	ctDNA is a good diagnosis method for early stage PDAC and for identification of cancer origin	20	63%	99%	panel of > 100,000 methylation regions

ctDNA, circulating tumour DNA; PC, pancreatic cancer; PDAC, pancreatic ductal adenocarcinoma.

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
