# Peer review of "Circulating Cell-Free Tumour DNA for Early Detection of Pancreatic Cancer"

_cancers, 2020, doi:10.3390/cancers12123704_

Round 1

Reviewer 1 Report

Very good work. Thank you for the possibility to review it.

Reviewer 2 Report

All my comments have been addressed

Reviewer 3 Report

Thank you for addressing the comments and revising the manuscript. It is sound, timely and well-written and I have no further questions.

This manuscript is a resubmission of an earlier submission. The following is a list of the peer review reports and author responses from that submission.

Round 1

Reviewer 1 Report

This is a well written and comprehensive review on the emerging field of cell free DNA in cancer.

There are only some minor remarks: 1. the introduction. Some of the statements are quite trivial and for western countries not right (5-year survival is higher then 5% i.e.). 2. futere/conclusion: maybe it would be better not to mention the companys but to show future possibilities.

Author Response

This is a well written and comprehensive review on the emerging field of cell free DNA in cancer.

Thank you for your positive comments.

There are only some minor remarks: 1. the introduction. Some of the statements are quite trivial and for western countries not right (5-year survival is higher then 5% i.e.).

We have updated the text, stating a 5-10% 5-year survival rate, which was obtained from the Cancer Research UK website (www.cancerresearchuk.org) and the NCI websites (https://seer.cancer.gov/statistics/).  In the UK, the 5 year survival is still hovering around 5% as stated on the CRUK website. The senior author has discussed this with the organisation and there are complex reasons why this is the case in the UK, we have not gone into this for the manuscript as it is outside the scope of the manuscript. Reasons for this are late presentation due to busy NHS service, inequity of oncology care due to access of teaching hospital vs community hospital.

Lines 40-44: Pancreatic ductal adenocarcinoma also has the lowest five-year survival rate of any malignancy, which is around 5-10%.

  1. future/conclusion: maybe it would be better not to mention the companys but to show future possibilities.

We have removed all company names and made the suggestions from the reviewer.

Lines 310-312: Newer approaches incorporate additional properties of ctDNA beyond sequencing, such as the length of fragments or methylation signatures, and combined with cancer-associated protein biomarkers.

Reviewer 2 Report

Comments to the Authors

The authors do a nice job summarizing the latest findings on development of circulating cell-free tumor DNA technology as it applies to pancreatic cancer. Overall, the opinions presented are balanced and cite relevant recent studies. There are some areas that could be discussed more thoroughly, some examples are provided below:

-It would be helpful for the reader to include discussion of how specificity and sensitivity can be determined in this assay and what would the acceptable metrics look like.

-Discussion on how the manner of DNA release (secretion vs cell death) may affect ctDNA detection

-Discussion on practicality of applying ctDNA technology for the purposes of early detection of pancreatic cancer, given the vague nature of early stage symptoms in this disease. What are the general considerations with regards to using ctDNA as a tool for screening the general population etc?   

Author Response

The authors do a nice job summarizing the latest findings on development of circulating cell-free tumor DNA technology as it applies to pancreatic cancer. Overall, the opinions presented are balanced and cite relevant recent studies.

Thank you for your positive comments.

There are some areas that could be discussed more thoroughly, some examples are provided below:

-It would be helpful for the reader to include discussion of how specificity and sensitivity can be determined in this assay and what would the acceptable metrics look like.

We agree that it would be helpful to discuss the methodology, particularly as it varied between the studies cited. Therefore, we have amended the text to include the discussion of the desired sensitivity and specificity calculation methods.

Lines 219-226: One challenge related to ctDNA biology is a lack of consistency in the study design. As investigated tumour stages vary between studies, which include both, case-control and cohort studies, a robust comparison of different methods is challenging. Moreover, PDAC should be confirmed using established techniques (e.g. histology), with a follow-up period for the detection of false-negative early-stage cancers. In addition, controls should consist of individuals in high-risk groups, likely to be frequently screened, as well as healthy volunteers. If possible, calculations of sensitivity and specificity should be applied to each tumour type and stage separately to enable reliable comparison between studies.

-Discussion on how the manner of DNA release (secretion vs cell death) may affect ctDNA detection

In line with Reviewer 2’s comment we have included the discussion of ctDNA origin, in particular of the ctDNA mode of release from cancer cells.

Lines 122-127: In patients diagnosed with cancer, cfDNA is also released from cancer cells, though the mechanism of ctDNA release remains unclear; either active secretion and/or release in cell death, supported by studies showing that ctDNA levels increase rapidly after treatment is administered in a range of cancers. The mechanism of DNA release might affect cancer detection, as actively secreted ctDNA may be biased in certain chemotherapy-sensitive metastatic sites.

-Discussion on practicality of applying ctDNA technology for the purposes of early detection of pancreatic cancer, given the vague nature of early stage symptoms in this disease. What are the general considerations with regards to using ctDNA as a tool for screening the general population etc?

Thank you for your suggestion. We agree that population screening for pancreatic cancer using ctDNA is unlikely due to a number of reasons e.g. cost, efficacy and incidence of this disease. We do recognise that the vague nature of the symptoms means that most patients are often diagnosed with the condition in an advanced stage where cure is highly unlikely. We have therefore made a number of suggestions regarding certain “at risk” groups of individual patients who may benefit from targeted screening.  

Lines 227-244: Efficient, early diagnosis of PDAC also requires the identification of the screening population, at risk of developing PDAC. There is a number of risk factors associated with PDAC, with the most relevant being new-onset diabetes mellitus, chronic pancreatitis or genetic predisposition. Ben et al. showed that while long-lasting diabetes mellitus increases the risk of developing PDAC by 1.5-fold, recent detection (within 1-3 years from the disease onset) increases this ratio to 5-8-fold. Similarly, the risk of developing PDAC increases 8-fold over the five-year period following chronic pancreatitis diagnosis. Finally, genetic predispositions is estimated to contribute to 5-10% of all PDAC cases, with mutations in 6 most significant genes being reported in 5.5% of patients. Altogether, currently available data points at certain risk factors, which should determine the screening population for PDAC using ctDNA-based liquid biopsy.

Reviewer 3 Report

In the present manuscript the authors provide an overview on the potential value of ctDNA for the early detection of pancreatic cancer. This cancer type is characterized by an extremely poor prognosis, most likely due to a late diagnosis in most cases with limited treatment options. ctDNA is shed by the tumor cells into the blood plasma and its amount correlates with tumor burden. The analysis of ctDNA may cover the heterogeneity of the tumor cells better than tissue biopsies might be able to, furthermore blood can be drawn at serial time points. On the other hand the value of ctDNA to detect early stages may be limited by the detection sensitivity of the applied analytics.

The manuscript reviews the literature on ctDNA in pancreatic cancer; however the titel of the manuscript conveys promises which, once they have been examined in detail, are unlikely to be fulfilled.

The major issues of criticism are:

  1. In line 122 the authors state that table 1 would list studies on early PDAC detection; however, there are also studies listed investigating ctDNA in all or even only late stages (e.g. Maire et al). Table 1 has to be updated, or at least the authors have to declare the selection criteria of the cited studies.
  2. List of references must be checked: e.g. in table 1 Maire et al and Dianxu et al are both [59], and the reference of Liu 2019 is missing. Likewise all references from chapter 7 on are not cited correctly, probably earlier reference as well.
  3. Structure of the article: I recommend to combine chapter 1 and 2 (Introduction); 3 and 4 (Liquid biopsies for ctDNA), because they do not focus in PDAC in particular.
  4. In line 147f The CancerSeek panel might be discussed in more detailed. There has been some controversies on that paper (see doi: 10.21037/jtd.2018.06.58) regarding the sensitivity of the repective cancer types.

Author Response

In the present manuscript the authors provide an overview on the potential value of ctDNA for the early detection of pancreatic cancer. This cancer type is characterized by an extremely poor prognosis, most likely due to a late diagnosis in most cases with limited treatment options. ctDNA is shed by the tumor cells into the blood plasma and its amount correlates with tumor burden. The analysis of ctDNA may cover the heterogeneity of the tumor cells better than tissue biopsies might be able to, furthermore blood can be drawn at serial time points. On the other hand, the value of ctDNA to detect early stages may be limited by the detection sensitivity of the applied analytics.

The major issues of criticism are:

In line 122 the authors state that table 1 would list studies on early PDAC detection; however, there are also studies listed investigating ctDNA in all or even only late stages (e.g. Maire et al). Table 1 has to be updated, or at least the authors have to declare the selection criteria of the cited studies.

We thank Reviewer 3 for their suggestions. To address the comment, we have (i) changed the title of Table 1 to indicate that we have included not only early detection studies, but also other articles describing the applicability of ctDNA for PDAC treatment, and (ii) indicated early PDAC studies with asterixis (*). We believe that other ctDNA PDAC studies are important to understand the progress over the years, and the feasibility of ctDNA use for early PDAC detection.

List of references must be checked: e.g. in table 1 Maire et al and Dianxu et al are both [59], and the reference of Liu 2019 is missing. Likewise, all references from chapter 7 on are not cited correctly, probably earlier reference as well.

We would like to thank the reviewer for picking up on this. We would like to profusely apologise for this error and it was an oversight on our part. We have checked all the references in the manuscript and corrected this accordingly.

Structure of the article: I recommend to combine chapter 1 and 2 (Introduction); 3 and 4 (Liquid biopsies for ctDNA), because they do not focus in PDAC in particular.

Thank you for your comments. We agree with the Reviewer 3 and have now combined Chapters 1 and 2 (entitled: Introduction) and Chapters 3 and 4 (entitled: Liquid biopsies for ctDNA).

In line 147f The CancerSeek panel might be discussed in more detailed. There has been some controversies on that paper (see doi: 10.21037/jtd.2018.06.58) regarding the sensitivity of the respective cancer types.

We agree with Reviewer 3’s comment regarding CancerSeek panel. We have amended the text to include the controversies of the methods used in the study, as described above.

Lines 209-218: However, the accuracy of the approach has been questioned. In particular, Young et al (2018) criticized the use of case-control data where subject selection and a choice of robust control is challenging. Specifically, no advanced stage cancers were included, and the healthy controls were likely to be biased and not represent the potential screening population (e.g. individuals with gastrointestinal tract symptoms or those characterised by genetic predispositions). In addition, the cases analysed included eight different cancer types of various stages, and it may therefore not be possible to draw robust conclusions about the utility of this approach in a single tumour type. For these reasons, the sensitivity and specificity reported could be biased and an additional study is required to validate the method.

Reviewer 4 Report

Jaworski et al. describe the use of ctDNA for early detection of pancreatic cancer in a review article. The article gives a good overview of the current status of ctDNA in PDAC. However, some major points should be addressed:

  • In the introduction you state that pancreatic cancer can only be cured by resection and adjuvant therapy. Recent advances in neoadjuvant therapy have shown to benefit even patients with locally advanced disease by making them suitable for curative resection. Neoadjuvant therapy may furthermore improve overall survival in borderline resectable and resectable patients. Please quote the latest evidence.
  • Line 58: What kind of liquid biopsy was that (blood, urine etc.)? At what stage of disease was it taken (locally advanced vs. metastastic)? Had treatment been administered in these patients (adjuvant therapy etc.)?
  • Line 86: Please provide more detail on the percentage ctDNA depending on stage as this is crucial to early cancer detection.
  • Please provide more details on specificity to differentiate ct DNA from cfDNA of benign cells.
  • How is the amount and quality of ct DNA impacted by chemo- or radiotherapy? That might be an important confounder if tracking disease progression.
  • Please provide more information on selection criteria for patients to undergo ct DNA early detection. Should it be performed in random individuals (probably very low sensitivity and specificity and not cost-effective) or in patients with symptoms or diagnostics suspicious for PDAC (newly diagnosed diabetes mellitus etc.)?
  • Are there studies available combining ct DNA marker panels with standard PDAC diagnostics (CT, EUS, biopsy..). How is diagnostic accuracy impacted?
  • Is there a difference in ct DNA detection for patients with nodal negative and nodal positive early PDAC? Would loco-regional invasion be mirrored in the amounts of cf tumor DNA?

Minor

  • I recommend calling paragraph 2 “Rationale for liquid biopsies in the detection of pancreatic cancer at an early stage”.

Author Response

Jaworski et al. describe the use of ctDNA for early detection of pancreatic cancer in a review article. The article gives a good overview of the current status of ctDNA in PDAC.

Thank you for your positive comments.

However, some major points should be addressed:

In the introduction you state that pancreatic cancer can only be cured by resection and adjuvant therapy. Recent advances in neoadjuvant therapy have shown to benefit even patients with locally advanced disease by making them suitable for curative resection. Neoadjuvant therapy may furthermore improve overall survival in borderline resectable and resectable patients. Please quote the latest evidence.

We have amended the text to include the latest evidence about the applicability of neoadjuvant therapy in borderline resectable and resectable patients.

Lines 46-49: Alternatively, the borderline resectable and locally advanced PDAC cases may benefit from neoadjuvant therapy, which can be safer than surgery. A number of studies have compared the two approaches but neoadjuvant therapy broad clinical utility still requires further investigation in a phase 3 clinical trial setting.

Line 58: What kind of liquid biopsy was that (blood, urine etc.)? At what stage of disease was it taken (locally advanced vs. metastastic)? Had treatment been administered in these patients (adjuvant therapy etc.)?

Thank you for your comment. Both liquid biopsies were blood based. Both papers used papers who had localised disease. Both papers had ctDNA measurement post-resection but pre-adjuvant treatment.

Line 86: Please provide more detail on the percentage ctDNA depending on stage as this is crucial to early cancer detection.

Thank you for your comment. We have included this information from the Bettegowda paper.

Lines 128-131: Moreover, Bettegowda et al. showed that early stage cancers have a lower percentage of ctDNA then late stage malignancies. For pancreatic cancer, ctDNA was detected in 48% of patients with localized tumours but in more than 80% of advanced cancers. This shows a potential limitation of ctDNA as an early detection tool for pancreatic cancer.

Please provide more details on specificity to differentiate ct DNA from cfDNA of benign cells.

Thank you for your suggestion. The core method used to differentiate cancerous versus non-cancerous DNA is the presence of variants in oncogenes/tumour suppressor genes that are either ubiquitously mutated in PDAC (e.g. KRAS) or are detected within the tumour DNA itself and then searched for within the ctDNA.

Lines 150-163: There is approximately 10-15 ng of cfDNA per millilitre and ctDNA constitutes a small fraction of cfDNA in most early stages of cancer. These factors highlight the need for ultrasensitive detection techniques. In clinical trials, current sequencing methods involve targeted or genome-wide approaches to genetic sequencing. The former is more prevalent and only detects hotspot mutations in specific genes. This technology could be applicable to somatic KRAS mutations ubiquitously found in PDAC. Conversely, genome-wide sequencing assays are an unrestricted method that is more expensive and technically challenging, but identifies many more mutations across the genome as well as structural variants and mutational signatures. Currently, the most sensitive methods are polymerase chain reaction (PCR)–based approaches, including single-molecule PCR BEAMing, TAm-Seq, digital PCR and droplet digital PCR. Next-generation sequencing techniques are also widely used, with the biggest limiting factor being the low fidelity of DNA polymerase. However, the incorporation of deep sequencing, molecular barcoding and digital error suppression can increase sensitivity and specificity of the approach.

How is the amount and quality of ct DNA impacted by chemo- or radiotherapy? That might be an important confounder if tracking disease progression.

The amount of ctDNA produced rises following cancer cell death from chemo/radiotherapy. In addition, the timing of the collection of samples to detect ctDNA is crucial in determining the most appropriate use of this technology e.g. a certain percentage increase/fold change in baseline levels of ctDNA and ctDNA detected within the first hours, days, week of giving chemo/radiotherapy may predict better objective responses (RECIST, CA199) to chemotherapy and may therefore be used to either truncate or elongate therapeutic regimens. Interestingly, Osumi et al. (2019) showed that low ctDNA levels 8 weeks after metastatic colorectal cancer chemotherapy were associated with longer PFS and OS.

Please provide more information on selection criteria for patients to undergo ct DNA early detection. Should it be performed in random individuals (probably very low sensitivity and specificity and not cost-effective) or in patients with symptoms or diagnostics suspicious for PDAC (newly diagnosed diabetes mellitus etc.)?

We agree with Reviewer 4’s comment. We have modified the manuscript in line with the Reviewer 4’s comment by suggesting targeted screening on patients with “at risk” conditions for the development of PDAC.

Lines 227-244: Efficient, early diagnosis of PDAC also requires the identification of the screening population, at risk of developing PDAC. There is a number of risk factors associated with PDAC, with the most relevant being new-onset diabetes mellitus, chronic pancreatitis or genetic predisposition. Ben et al. showed that while long-lasting diabetes mellitus increases the risk of developing PDAC by 1.5-fold, recent detection (within 1-3 years from the disease onset) increases this ratio to 5-8-fold. Similarly, the risk of developing PDAC increases 8-fold over the five-year period following chronic pancreatitis diagnosis. Finally, genetic predispositions is estimated to contribute to 5-10% of all PDAC cases, with mutations in 6 most significant genes being reported in 5.5% of patients. Altogether, currently available data points at certain risk factors, which should determine the screening population for PDAC using ctDNA-based liquid biopsy.

Are there studies available combining ct DNA marker panels with standard PDAC diagnostics (CT, EUS, biopsy..). How is diagnostic accuracy impacted?

The studies cited combined ctDNA analysis with protein biomarkers (e.g. CA19-9) to increase sensitivity and specificity of the approach. Other detection methods (e.g. CT, EUS, histology) were used to identify true-positives and true-negatives, instead. We are not aware of studies incorporating standard diagnostic methods with ctDNA.

Is there a difference in ct DNA detection for patients with nodal negative and nodal positive early PDAC? Would loco-regional invasion be mirrored in the amounts of cf tumor DNA?

We are not aware of studies comparing ctDNA in nodal positive and nodal negative PDAC. We can hypothesise that indeed, tumour progression to the lymph nodes may increase the amount of ctDNA in blood but this remains to be investigated.

Minor

I recommend calling paragraph 2 “Rationale for liquid biopsies in the detection of pancreatic cancer at an early stage”.

Thank you for your suggestion, we have changed the structure of the manuscript in line with Reviewer 3’s comments. As a result, paragraphs 1 and 2 have been combined, and we hope that this addresses Reviewer 4’s comment.